# Trend and Joinpoint Analysis of Cancer Incidence and 1-Year Mortality in North-East Spain 2005–2020

**DOI:** 10.3390/cancers15235527

**Published:** 2023-11-22

**Authors:** Pere Roura, Emma Puigoriol, Jacint Altimiras, Eduard Batiste-Alentorn, Irene R. Dégano

**Affiliations:** 1Clinical Epidemiology and Research Unit, Vic Hospital Consortium, 08500 Vic, Spain; pere.roura@umedicina.cat (P.R.); epuigoriol@chv.cat (E.P.); jaltimiras@chv.cat (J.A.); 2Faculty of Medicine, University of Vic—Central University of Catalonia, 08500 Vic, Spain; 3Institute for Research and Innovation in Life Sciences and Health in Central Catalonia (IRIS-CC), 08500 Vic, Spain; 4Clinical Oncology Department, Vic Hospital Consortium, 08500 Vic, Spain; ebalentorn@hotmail.com; 5Centro de Investigación Biomédica en Red of Cardiovascular Diseases (CIBERCV), Instituto de Salud Carlos III, 28029 Madrid, Spain; 6Registre Gironí del Cor (REGICOR) Study Group, Hospital del Mar Medical Research Institute (IMIM), 08003 Barcelona, Spain

**Keywords:** neoplasms, incidence, mortality, trends, regression analysis, registries, COVID-19

## Abstract

**Simple Summary:**

There are scarce data on 1-year mortality cancer trends and on the effect of the COVID-19 pandemic on cancer occurrence and mortality. In addition, there is no information on cancer trends for several European regions. We studied cancer occurrence and 1-year mortality during 2005–2020 in Osona, a North-Eastern Spanish region without previously reported data. We found that the occurrence of colorectal, lung and bronchus, and urinary bladder cancer increased in females, while the occurrence of colorectal and prostate cancer decreased in males. Moreover, 1-year mortality decreased for endometrium and for male colorectal cancer. Our results show differences with other Spanish and European regions, pointing to the need of analyzing cancer trends in as many areas as possible to improve cancer management. Finally, from 2019 to 2020, cancer occurrence decreased and 1-year mortality of cancer patients increased. This result suggests that plans should be designed to avoid cancer diagnosis and treatment delays when the healthcare system is collapsed.

**Abstract:**

Cancer is the second leading cause of death. It is thus essential to examine cancer trends in all regions. In addition, trend data after 2019 and on cancer 1-year mortality are scarce. Our aim was to analyze incidence and 1-year mortality cancer trends in northeastern Spain during 2005–2020. We used the Osona Tumor Registry, which registers cancer incidence and mortality in Osona. The mortality information came from the Spanish Death Index. We analyzed age-standardized incidence rates and 1-year mortality by sex in the population aged > 17 years during 2005–2020. Trends were examined with negative binomial and joinpoint regression. Incidence rates of colorectal, lung and bronchus, and urinary bladder cancer increased annually in females by 2.86%, 4.20%, and 4.56%, respectively. In males, the incidence of stomach and prostate cancer decreased annually by 3.66% and 2.05%, respectively. One-year mortality trends decreased annually for endometrium cancer (−9.0%) and for colorectal cancer in males (−3.1%). From 2019 to 2020, the incidence of cancer decreased, while 1-year mortality increased in both sexes. In a North-Eastern Spanish county, 1-year mortality decreased for endometrium cancer in females and for colorectal cancer in males. Our results suggest a trend of decreasing cancer incidence and increasing cancer mortality as a result of the COVID-19 pandemic.

## 1. Introduction

Neoplasms are the second global cause of death and years of life lost worldwide [1,2]. In 2019, neoplasms caused 10 million deaths and 241 million years of life lost. By 2040, it is expected that these numbers will rise to 15 million and 296 million, respectively [3].

Taking into account the expected increase in cancer burden, it is mandatory to analyze cancer trends at local, regional, and global levels. Because cancer incidence, cancer mortality and cancer survival trends, are essential to organize initiatives, and health resources aimed at preventing, diagnosing and treating cancer [2]. However, only 45% of European inhabitants are covered by consolidated population-based cancer registries [4,5,6,7].

The most recent data on cancer incidence and mortality in Europe showed that the most common diagnosed cancers in females were breast, colorectal, and endometrium cancers. In males, these were prostate, lung, and colorectal cancers [8]. In recent decades, the mortality has decreased for most cancers, while the incidence trends depend on the cancer type and region. In a study of female cancers in Nordic countries, the incidence of endometrium cancer decreased, but breast cancer incidence trends differed between countries. In addition, the mortality of breast cancer decreased, and it decreased/stabilized for endometrium cancer [9]. In a study of seven high-income countries including Denmark, Ireland, Norway, and UK, incidence and mortality decreased for colorectal, lung (males), ovarian, and stomach cancer [10]. In France, incidence decreased the most for stomach cancer in females, and for larynx cancer in males. While it increased the most for lung cancer in females, and for thyroid cancer in males [11]. As for mortality, the largest decrease was observed in thyroid cancer in females, and in larynx cancer in males. While mortality increased the most for lung cancer in females, and for skin melanoma in males [11]. In Italy, the mortality rates decreased for most cancer types including breast, colorectal, prostate, stomach and urinary bladder cancers [12].

In Spain, there are several consolidated population-based registries [13], but the whole country is far from being covered. In regions without consolidated population-based registries, well-designed emergent population-based registries could be useful to examine cancer trends. Catalonia, in North-East Spain, has two consolidated population-based registries, but many reference hospitals are not included in these registries.

In addition, most population-based registries tend to examine the incidence and mortality at the population level, but not short-term mortality. As a result, trends on 1-year cancer mortality are not available for most cancer types and regions. Finally, cancer trends can also provide insight into cancer diagnosis and treatment in complex healthcare situations such as the COVID-19 pandemic [14]. But international cancer consortiums and population-based registries have not published much data on the effect of the COVID-19 pandemic on cancer trends so far.

The goal of this study was to examine trends in cancer incidence and 1-year mortality in Osona county during 2005–2020. The secondary goal was to examine the impact of the COVID-19 pandemic on cancer incidence and 1-year mortality trends.

## 2. Materials and Methods

### 2.1. Study Population and Data

The study population includes the inhabitants of the Osona County aged ≥ 18 years.

Osona County, in Catalonia, North-East Spain, has a reference hospital consortium and a population of 163,667 inhabitants. Osona’s hospital consortium developed the Osona Tumor Registry, a standardized cancer registry based on the recommendations of the European Network of cancer Registries (ENCR) [5], which has been active since 2005.

The data on cancer incidence from 2005 to 2020 were obtained from the Osona Tumor Registry and data on cases mortality was obtained from the Osona Tumor Registry and the Spanish National Death Index.

### 2.2. Osona Tumor Registry

The Osona Tumor Registry includes all cancer cases diagnosed or treated at the Vic Hospital Consortium, which incorporates all hospitals of the Osona County. The Vic Hospital Consortium includes the Vic General Hospital, the Vic Holy Cross Hospital, and Saint James Hospital in Manlleu. The Vic General Hospital is the basic and community reference hospital, and the other two centers provide palliative, rehabilitation and social care for all residents in the Osona county. On the other hand, the Consortium is also the main provider of private healthcare in the county. Thus, the Osona Tumor Registry is able to identify all cancer cases diagnosed or treated in the Osona County.

The Osona Tumor Registry is a cancer registry that has been defined, organized and integrated within the healthcare information system (computerized clinical history) of the Consortium. The Osona Tumor Registry follows the recommendations of the ENCR regarding the standard dataset, the basis of diagnosis, and the definition of the incidence date among others. Thus, all of its operating procedures, data management and quality control to ensure registry comparability, validity and completeness are of a population-based registry. The use of international recommendations for the registration of cases and the procedures implemented on a well-defined epidemiological catchment area ensure that its data can be analyzed as the data from a consolidated population-based cancer registry.

### 2.3. Registration in the Osona Tumor Registry

The individual health identity card code is used to avoid duplicate patients and to confirm multiple cancers in a patient. The following clinical information is collected from each patient: date of diagnosis, that is, date of case incidence, center of diagnosis, tumor topography and cellular morphology according to ICD-O-3 [15], stage (0-IV and TNM system) [16], size and extent of main tumor, number of lymph nodes affected, presence of metastasis, first and adjuvant treatments, cancer relapse (presence, localization, and date), and vital status. All this information is collected as defined by international recommendations [5]. Vital status is obtained from the Vic Hospital Consortium’s clinical records when the patient dies in one of the included hospitals or from the Spanish National Death Index (Spanish Ministry of Health); otherwise [17].

### 2.4. Events of Interest

The incidence rates and mortality at 1-year were calculated from 2005 to 2020 for the age group ≥18 years, by year, sex, and tumor localization. As for tumor localization, we selected all cancers together as well as the most common cancer diagnoses in Osona (colorectal, breast, prostate, lung, urinary bladder, stomach, and endometrium). These cancers include the most common diagnosed cancers in females and males reported from the last European cancer statistics [8]. Population denominators for the incidence rate calculation were obtained from the Statistical Institute of Catalonia [18]. Incidence rates were age-standardized using the 2013 European Standard Population [19]. While 1-year mortality was age-standardized using weights from the Surveillance, Epidemiology, and End Results Program (SEER), of the US National Cancer Institute [20]. Using the European standard population and standardized population data from SEER we warrant the comparability of our figures and results.

### 2.5. Statistical Analysis

Graphs of standardized incidence and 1-year mortality were presented with the values as dots, as well as, with a smooth line trend to improve interpretation due to the high variability. The line trend was computed with smooth local regression (loess).

Trends in standardized incidence rates and 1-year mortality for the period 2005–2020 were examined with negative binomial regression (glm.nb R function). In these regression models, the year was the independent variable. We calculated the annual percentage change (APC) as the regression coefficient of year × 100, and its 95% confidence interval (CI). Negative binomial results are presented with forest plots. Negative binomial assumptions of linearity in model parameters and independence of individual observations were met. Linearity was assessed graphically by plotting the expected counts across the range of years. Independence was examined with the Durbin-Watson test.

Trends were also analyzed with joinpoint regression to identify significant changes in trends. To define the joinpoint models we set as 2 the minimum number of years between joinpoints and from a joinpoint to either end of the data. We also set the maximum number of joinpoints as 2. Joinpoint models were fitted as log-linear models assuming uncorrelated errors. The best model was selected using the permutation test. First, the residuals from a model with no joinpoints were permuted. Then, the permutation test was used to define how large the ratio of the sum of squared errors from 2 models (with and without joinpoints) had to be for identification as statistically significant. More information on the permutation test can be found in Appendix A. APCs and CIs were obtained for all trend periods with significant joinpoints.

We performed sensitivity analyses to corroborate the identified joinpoints and to analyze the impact of autocorrelation in the joinpoint regression. To corroborate the joinpoints we performed an analysis with generalized linear models (glm and segmented R functions) and calculated whether the regression parameter of year was constant using the Davies test (davies.test R function). As for the impact of autocorrelation, we conducted an analysis assuming first order autocorrelated errors with correlation 0.1, 0.2 and 0.3.

Negative binomial regression and sensitivity analyses to corroborate joinpoints were performed with R Statistical Software version 4.2.1 [21]. Joinpoint regression and sensitivity analyses for autocorrelation were conducted with the Joinpoint Regression Program (National Cancer Institute Bethesda, MD, USA, version 4.9.0) [22,23].

## 3. Results

### 3.1. Number of Cases

From 2005 to 2020 there were 11,434 Osona inhabitants diagnosed with cancer. Seven cancer sites represented 71% of the cases: colorectal (19%), breast (15%), prostate (12%), lung and bronchus (9%), urinary bladder (9%), stomach (4%) and endometrium (3%).

### 3.2. Incidence Rates

Crude incidence rates are presented in Appendix A. Standardized incidence rates of any cancer decreased in both sexes (600-497/100,000), but the decrease was larger in males (836-621/100,000) than in females (440-405/100,000) (Appendix A, Figure 1). In females, the highest incidence rates were observed for breast (189-131/100,000) and colorectal cancer (51-63/100,000) (Appendix A and Figure 2A). While in males, the highest rates were found for prostate (218-125/100,000) and colorectal (157-130/100,000) cancer (Appendix A and Figure 2B).

### 3.3. Incidence Rate Trends

According to incidence estimates in both sexes together, the largest changes during 2005–2020 were a decrease in the incidence of all cancer cases and stomach cancer, as well as an increase in the incidence of bladder cancer (Appendix A). However, the variation in the incidence rates was large for all cancers and for bladder cancer. In females, the largest changes were observed as a decrease of all cancers, breast, and stomach cancer (Appendix A and Figure 1, Figure 2A and Appendix A). But again, these trends showed great variation. On the other hand, there was an increase in colorectal, lung, and bladder cancer with a more robust pattern. In males, the largest changes were a decrease in incidence of all cancers and of prostate cancer (Appendix A and Figure 1 and Figure 2B). In addition, there was a clear decreasing trend for stomach cancer incidence.

The results from the negative binomial regression models showed that incidence rates of colorectal, lung and bronchus, and urinary bladder cancer increased in females by 2.86%, 4.20% and by 4.56% annually, respectively (Figure 3A). While in males, the incidence of any cancer, stomach, and prostate cancer decreased annually by 0.92%, 3.66%, and 2.05%, respectively.

The joinpoint analyses showed a decrease in all-cancer incidence in the last 5 years in females (6.9% annually) and in males (8.8% annually) (Table 1). In females, there was a previous increase in all cancer incidences (2005–2016, 2.3% annually). Regarding specific cancer sites, the urinary bladder cancer incidence increased in 2005–2009 in males (a 23.6% annual increase) and stabilized thereafter. In females, there was a change in the breast incidence trend in 2017.

### 3.4. 1-Year Mortality

Standardized 1-year mortality for all cancers in 2020 was 24%, 19% in females, and 29% in males (Appendix A). In 2020, the highest 1-year mortality was observed in stomach and lung cancer in females (47% and 42%, respectively) and in males (72% and 66%, respectively). These were also the cancer localizations showing the largest 1-year mortality during the whole period (Figure 4).

During 2005–2020, as shown by standardized mortality data, 1-year mortality in females and males together halved for colorectal cancer (24 to 12%) and doubled for stomach cancer (27 to 64%) (Appendix A). However, 1-year mortality for stomach cancer had much variation. All cancer trends were similar in females and males (Figure 5). In females, 1-year mortality increased for stomach cancer (29 to 47%) and decreased for bladder cancer (25 to 5%) and endometrium cancer (18 to 3%) (Appendix A and Figure 4A and Appendix A). In addition, endometrium cancer 1-year mortality showed the most consistent trend. In males, 1-year mortality increased for stomach cancer (26 to 72%) and decreased for colorectal cancer (25 to 8%), with the latter showing the most robust trend (Appendix A and Figure 4B).

Negative binomial 1-year mortality trends decreased significantly for endometrium cancer in females (−9.0% annually), as well as for colorectal cancer in males (−3.4% annually), and in females and males together (−3.1% annually) (Figure 3B). 1-year mortality trends showed a joinpoint for lung and bronchus cancer in males and in females and males together (Table 1). In males, there were 2 joinpoints: in 2015 and 2018, 1-year mortality increased in the first period (1.3% annually), decreased in the second (−13.7% annually), and remained stable in the third.

### 3.5. Incidence and 1-Year Mortality of Cancer during the COVID-19 Pandemic

In 2020, the incidence of all cancers decreased in both females and males, while 1-year mortality of all cancers increased (Figure 1 and Figure 5, and Appendix A). Regarding cancer types, there was a decrease in the incidence of breast, colorectal, and stomach cancers in females (Figure 2A and Appendix A, and Appendix A). While in males, there was a decrease in the incidence of bladder, colorectal, lung and bronchus, prostate, and stomach cancers (Figure 2B and Appendix A). 1-year mortality increased in 2020 for colorectal and stomach cancer in females (Figure 4A and Appendix A), and for bladder, lung and bronchus, prostate, and stomach cancer in males (Figure 4B and Appendix A).

### 3.6. Sensitivity Analyses

Sensitivity analyses provided similar results when assuming autocorrelation and regarding the identified joinpoints (Appendix A).

## 4. Discussion

During 2005–2020 in Osona county, North-East Spain, the cancer incidence and 1-year mortality was higher in males than in females. The highest incidence was observed for breast and colorectal cancer in females, and for colorectal and prostate cancer in males. In this period, there was an increase of bladder, colorectal, and lung and bronchus cancer incidence in females. While in males, the incidence of prostate and stomach cancer decreased. In both sexes, the highest 1-year mortality was found for lung and bronchus, and for stomach cancer. During 2005–2020, 1-year mortality decreased for endometrium cancer in females and for colorectal cancer in males. As for the effect of the COVID-19 pandemic on cancer trends, the incidence of all cancers was lower in 2020 than in 2019, whereas 1-year mortality was higher. Only the endometrium, and lung and bronchus cancer incidence in females increased in 2020. Regarding 1-year mortality, lower figures were only observed in 2020 for bladder, breast, and lung and bronchus cancer in females, and for colorectal cancer in males.

### 4.1. Trends in Cancer Incidence

Trends on cancer incidence in Osona displayed an increase in colorectal, lung and bronchus, and urinary bladder cancer in females. While in males, there was a decrease in prostate and stomach cancer incidence. GLOBOCAN data until 2010–2016 presents an increasing trend for colorectal, lung and bronchus, and urinary bladder cancer incidence in females during the last years and decades in 55%, 65%, and 33% of the 42 available countries in the world [24]. As for worldwide trends in males, in 71% and 36% of the 42 GLOBOCAN countries, there were decreasing trends in prostate and stomach cancer, respectively.

In Spain and Catalonia, during 2012–2022, there were increasing trends for colorectal, lung and bronchus, and urinary bladder cancer incidence in females [25]. In males, the prostate cancer incidence increased during this period, while the stomach cancer incidence decreased until 2020 and increased thereafter. Our results on female incidence trends are in agreement with recent trend data from Spain and Catalonia. However, we found a decrease in prostate cancer incidence in males during 2005–2020, while an increase was observed in Spain and Catalonia. This discrepancy as well as the differences observed in GLOBOCAN data by country highlight the dissimilarities that exist between regions. These dissimilarities emphasize the need for cancer surveillance in as many regions as possible to define prevention and management plans that are meaningful for each region.

The increase in colorectal, lung and bronchus, and urinary bladder cancer incidence in females could be associated with several factors. On the one hand, this rise could be related to cancer risk factors such as smoking and obesity [26]. The prevalence of daily smokers decreased in Spain during 2009–2020. However, the decrease was larger in males than in females [27]. In addition, the prevalence of overweight and obesity increased in Spain during 1987–2017, and this increase was larger in females than in males [28]. On the other hand, screening programs could have also contributed to the increase in the incidence of certain cancers, for example, colorectal cancer. The Spanish program for colorectal cancer screening started in 2014 [29].

### 4.2. Trends in Cancer Mortality

1-year mortality trends in Osona decreased for endometrium cancer in females and for colorectal cancer in males. While there is no published data on trends of 1-year mortality in these cancers in global collaborations or in Spain, there is data regarding annual mortality due to these cancers. Mortality analysis of GLOBOCAN data from 46 countries until 2015 approximately, showed a reduction in colorectal cancer mortality in males in 65% of the countries, but a reduction in endometrium cancer mortality in females in only 16% of the countries [24]. Data from Catalonia during 1993–2007 showed a stabilization of colorectal cancer mortality in males and a reduction of endometrium cancer mortality in females [25]. Observed discrepancies are probably caused by the different outcomes analyzed (1-year vs. annual mortality).

Reduction of colorectal cancer 1-year mortality could be linked to expansion of screening programs and to improvement of cancer survival. The implementation of the colorectal cancer screening program in Spain increased the screening coverage of the population during 2009–2017 from 3 to 22% [28]. In Osona, colorectal cancer screening started in 2015 and has a coverage of 60%. This increase in screening has probably raised the diagnosis of colorectal cancers in I-III stages, which are associated with lower 1-year mortality [30,31]. As for cancer survival, a reduction in colorectal cancer 1-year mortality in patients with stage I-III colon was observed in the Netherlands during 2009–2013 [32]. In addition, 5-year survival of colorectal cancer increased significantly in Spain in the population aged < 75 years [33].

### 4.3. Incidence and 1-Year Mortality of Cancer during the COVID-19 Pandemic

During March–June 2020 there were huge reductions in cancer referrals, gold standard tests, chemotherapy attendance, and patients receiving surgery [34,35]. In addition, cancer screening was lower after the lockdown compared to the period before the pandemic [36]. The reduction in diagnostic tests suggests that a lower number of individuals received a cancer diagnosis during 2020 [35]. This reduction in cancer diagnosis together with the decrease of cancer treatments could increase cancer mortality from 2020 on [34,37].

Our results showed that in Osona, the overall cancer incidence decreased from 2019 to 2020 in both females and males. In Spain from 2019 to 2020 the cancer incidence decreased in 30% of cancer types in females and males, while in 2020–2022 it decreased in 7% and 11% of cancer types in females and males, respectively [38], showing a more pronounced decrease of cancer incidence in 2020.

On the other hand, the overall cancer 1-year mortality increased in Osona from 2019 to 2020 in both females and males. In Spain, from 2019 to 2020, cancer mortality increased in 45% and 42% of cancer types in females and males, respectively. In addition, during 2020–2022 cancer mortality increased at least in 1 year in all cancer types, in both females and males [38], suggesting a higher cancer mortality after the first year of the COVID-19 pandemic.

### 4.4. Strengths and Limitations

This study has several strengths. First, it provides trend data on 1-year cancer mortality, which has been analyzed in very few previous studies. Second, it undertakes a joinpoint analysis of 1-year cancer mortality and of cancer incidence in a North-Eastern Spanish region. And third, it used high-quality data collected from a standardized cancer registry defined within a fixed epidemiological catchment area. The Osona Tumor Registry uses standardized methods for data collection and incorporates mortality from internal and external sources. In addition, the proportion of microscopically verified cancers in the Osona Tumor Registry is 91.4%, similar to the value reported by the two consolidated population-based cancer registries in Catalonia (Girona 89.4%; Tarragona 90.5%, [6]).

On the other hand, this study also has limitations that should be mentioned. Although the registry has a known population denominator and is located in the reference hospital consortium where cancer diagnosis and treatment are performed, it is possible that some Osona patients receive the diagnosis or treatment at third-level hospitals outside the county. Then, these patients would not be included in the registry at the time of incidence or treatment. However, these patients would be registered during the course of the disease, in case of hospital admission or referral to a palliative care unit, both in Osona county. Another limitation could be the presence of errors in the data or missing information. This limitation is minimized by the quality control of the data performed by trained personnel, as well as by the linkage of the registry with the computerized healthcare information system. This linkage avoids limitations from errors in the coding of disease. Moreover, the compulsory use of the health care identity card avoids case duplicates. To minimize missing values, data managers carry out daily checks to maintain and update data, and, if needed, they review the medical record to ensure the completeness of the information.

## 5. Conclusions

In Osona, North-Eastern Spain, during 2005–2020, we found an increase in colorectal, lung and bronchus, and urinary bladder cancer incidence in females. Additionally, we found a decrease in prostate and stomach cancer incidence in males. During the same period, we also observed a decrease in 1-year mortality for endometrium cancer in females and for colorectal cancer in males. Finally, from 2019 to 2020, there was a decrease in all cancer incidence and an increase in all cancer 1-year mortality. Importantly, our incidence trend results are in agreement with national and regional population-based cancer registries.

## Figures and Tables

**Figure 1 cancers-15-05527-f001:**
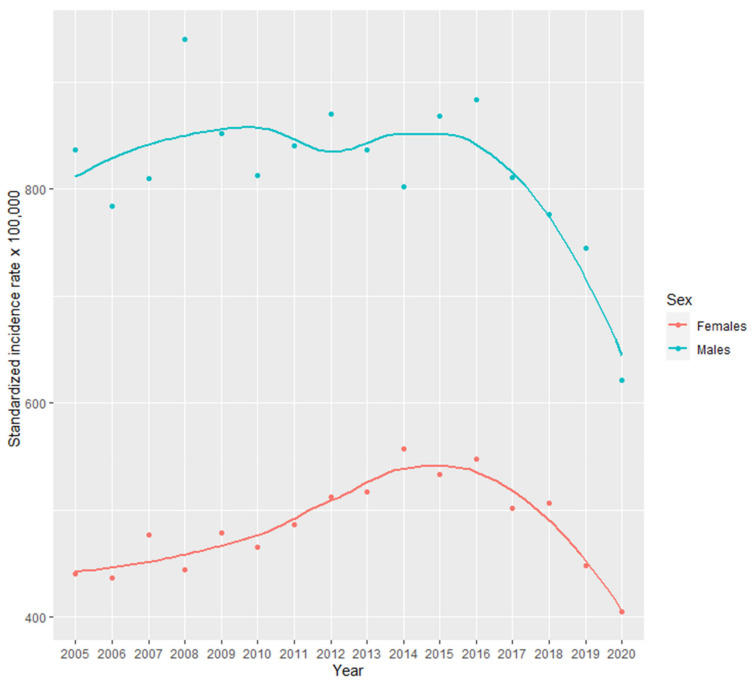
Cancer age-standardized incidence by sex in Osona during 2005–2020. Values are presented as dots. Trends are shown with a smooth local regression line. For age-standardization we used the 2013 European Standard Population.

**Figure 2 cancers-15-05527-f002:**
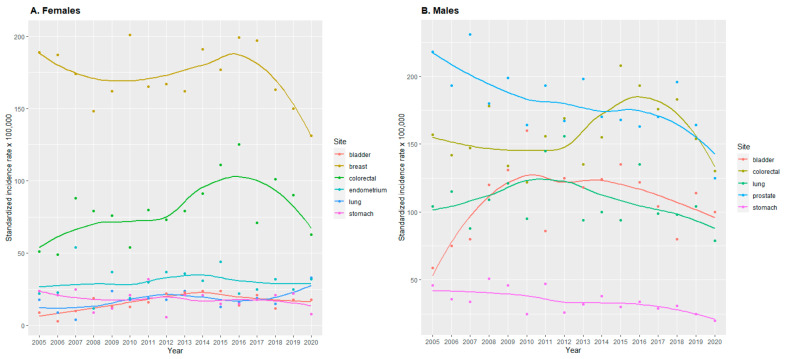
Cancer age-standardized incidence by cancer localization in females (**A**) and males (**B**) from Osona during 2005–2020. Values are presented as dots. Trends are shown with a smooth local regression line. For age-standardization we used the 2013 European Standard Population.

**Figure 3 cancers-15-05527-f003:**
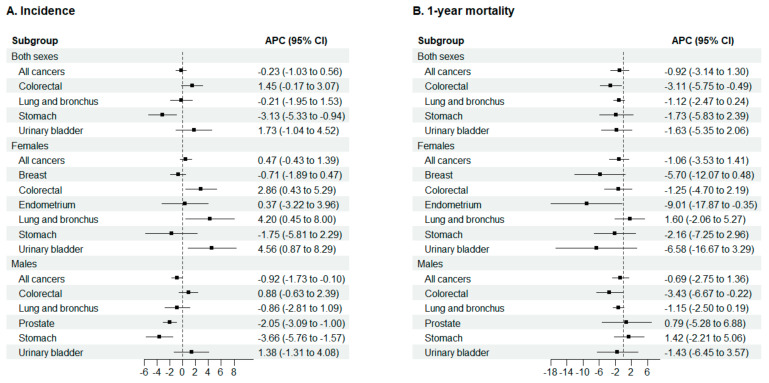
Trends in cancer age-standardized incidence (**A**) and 1-year mortality (**B**) by localization and sex in Osona during 2005–2020. All period trends were obtained with negative binomial regression. APC, Annual percentage change; CI, Confidence interval.

**Figure 4 cancers-15-05527-f004:**
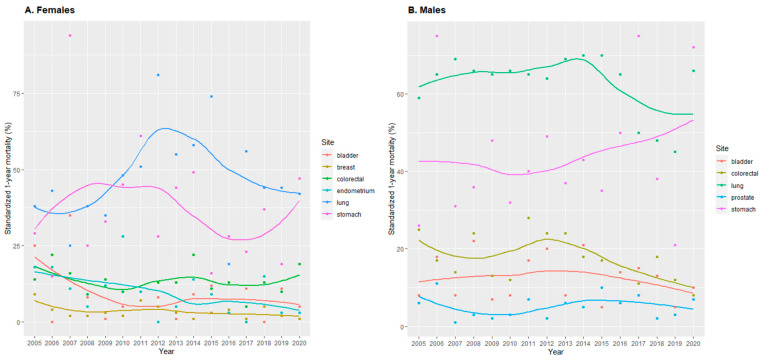
Cancer age-standardized 1-year mortality by cancer localization in females (**A**) and males (**B**) from Osona during 2005–2020. Values are presented as dots. Trends are shown with a smooth local regression line. For age-standardization we used the survival weights from the Surveillance, Epidemiology, and End Results Program of the US National Cancer Institute.

**Figure 5 cancers-15-05527-f005:**
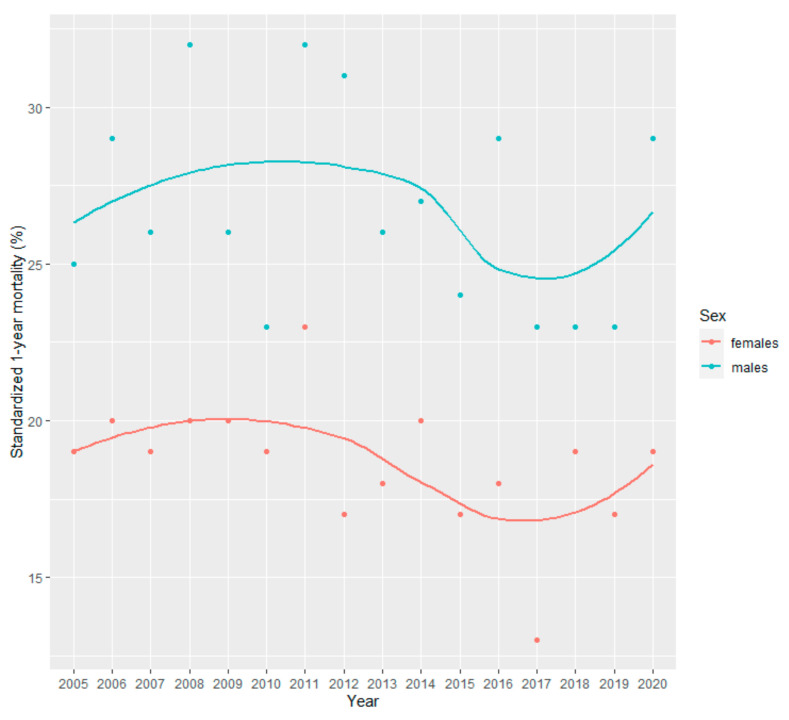
Cancer age-standardized 1-year mortality by sex in Osona during 2005–2020. Values are presented as dots. Trends are shown with a smooth local regression line. For age-standardization we used the survival weights from the Surveillance, Epidemiology, and End Results Program of the US National Cancer Institute.

**Table 1 cancers-15-05527-t001:** Joinpoint analysis of cancer standardized incidence and 1-year mortality in Osona during 2005–2020.

	Period	APC (95% CI)	Period	APC (95% CI)	Period	APC (95% CI)
**Incidence**						
**Both sexes**						
All cancers	2005–2016	1.3 (0.6, 2.0)	2016–2020	−**6.7** (−**9.6**, −**3.6**)		
Urinary bladder	2005–2009	**25.3** (**4.2**, **50.5**)	2009–2020	−2.6 (−5.7, 0.5)		
**Females**						
All cancers	2005–2016	**2.3** (**1.5**, **3.0**)	2016–2020	−**6.9** (−**10.1**, −**3.7**)		
Breast	2005–2017	0.6 (−1.0, 2.3)	2017–2020	−10.8 (−23.1, 3.4)		
**Males**						
All cancers	2005–2017	0.2 (−0.8, 1.1)	2017–2020	−**8.8** (−**16.0**, −**1.1**)		
Urinary bladder	2005–2009	**23.6** (**1.8**, **49.9**)	2009–2020	−2.8 (−6.0, 0.5)		
**1-year mortality**						
**Both sexes**						
Lung and bronchus	2005–2015	1.0 (−1.6, 3.7)	2015–2020	−6.9 (−13.6, 0.3)		
**Males**						
Lung and bronchus	2005–2015	**1.3** (**0.2**, **2.4**)	2015–2018	−**13.7** (−**24.3**, −**1.7**)	2018–2020	15.0 (−1.9, 34.9)

Only localizations with significant joinpoints are shown. Significant trends are presented in bold. Analyses were conducted with the Joinpoint Regression Program. APC: annual percentage change; CI: confidence interval.

## Data Availability

The data presented in this study are available on request from the corresponding author. The data are not publicly available because the data can only be accessed by the Osona Tumor Registry personnel.

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
