# Peer review of "Trend and Joinpoint Analysis of Cancer Incidence and 1-Year Mortality in North-East Spain 2005–2020"

_cancers, 2023, doi:10.3390/cancers15235527_

Round 1

Reviewer 1 Report

Comments and Suggestions for Authors

Author from Spain investigated incidence and 1-year mortality cancer trends in North-East Spain during 2005-2020. Incidence rates of colorectal, lung and bronchus, and urinary bladder cancer increased in females. In males, incidence of stomach and prostate cancer decreased annually. One-year mortality trends decreased annually for endometrium cancer and for colorectal cancer in males . From 2019 to 2020 incidence of cancer decreased while 1-year mortality increased in both sexes.

At the first sight, this is a good epidemiological work. However, there are some issues:

1.       Introduction includes a lot of information on methods (registries) which belongs to method chapter. Instead, introduction should mention other publications to this topic, probably from other European countries. What is known about this topic?

2.       Authors show decreasing or increasing trends using regression models. However, when looking at figures, these trends are difficult to recognize in many cases. Incidence rates jump up and down and do not have an linear trend. This is big limitation of this study, as readers have hard to believe the results only based on regression analyses. In usual studies, readers first see descriptive changes which then are confirmed by statistical methods. In the present study, there are no descriptive changes, but only statistical values. Either authors did not present results in the way that readers can understand them, or they should better work on their Figures in order readers to see changes descriptively before  complex statistics confirm them.

3.       Authors should better present study limitations; there are more limitations than only 1-2 they mentioned. Think about the registers itself; they always have challenges with missing data, code mistakes and so on.

Author Response

Author from Spain investigated incidence and 1-year mortality cancer trends in North-East Spain during 2005-2020. Incidence rates of colorectal, lung and bronchus, and urinary bladder cancer increased in females. In males, incidence of stomach and prostate cancer decreased annually. One-year mortality trends decreased annually for endometrium cancer and for colorectal cancer in males. From 2019 to 2020 incidence of cancer decreased while 1-year mortality increased in both sexes.

At the first sight, this is a good epidemiological work. However, there are some issues:

We want to thank the referee for the time devoted to review our manuscript, and for the comments, which have improved the manuscript. In the following paragraphs, we provide a point-by-point response to referee comments.

  1. Introduction includes a lot of information on methods (registries) which belongs to method chapter. Instead, introduction should mention other publications to this topic, probably from other European countries. What is known about this topic?

As the reviewer says, we talk about characteristics of registries in the introduction. We did so to place the beginning of interest in the clinical and epidemiological data of cancer and its registration, to describe some gaps that lead to the objectives of our study, and to contextualize our results. However, as the reviewer points out, there is a lack of information on what is already known about the topic. To improve the introduction we have reorganized the information that is presented, we have reduced part of the text regarding characteristics of cancer registries, we have moved part of the text to the methods section, and we have included a paragraph on the state of the art of incidence and mortality trends of cancer in Europe. The changes are marked in yellow.

  1. Authors show decreasing or increasing trends using regression models. However, when looking at figures, these trends are difficult to recognize in many cases. Incidence rates jump up and down and do not have an linear trend. This is big limitation of this study, as readers have hard to believe the results only based on regression analyses. In usual studies, readers first see descriptive changes which then are confirmed by statistical methods. In the present study, there are no descriptive changes, but only statistical values. Either authors did not present results in the way that readers can understand them, or they should better work on their Figures in order readers to see changes descriptively before complex statistics confirm them.

We totally agree with the referee comment. Incidence and mortality figures have substantial variation and it is difficult to see the trends in the graphs. In addition, we do not describe in the text many results of trends based on incidence and mortality data. We did not include much information on descriptive trends because we thought the text was too long, and we chose to focus on the results from the models. But we think now that these information is needed.

In order to improve the understanding of the results we have modified the figures to show a smooth line together with the observations as dots. Doing so, the trends are observed more clearly and with less variation. When needed, we have provided supplementary figures for certain cancer localizations if several trends were very close. In addition, we have added Table 1, and new figures in supplementary data (incidence and mortality for females and males together). On the other hand, figures have been rearranged so they are in the same order as the description of the results. Finally, we have included an explanation of descriptive changes in incidence and mortality in the results section and we have added an explanation of the smooth lines in the methods section. All these changes are marked in yellow in the manuscript.  

  1. Authors should better present study limitations; there are more limitations than only 1-2 they mentioned. Think about the registers itself; they always have challenges with missing data, code mistakes and so on.

As the reviewer mentions, the Osona Tumor Registry is sensitive to code mistakes and missing data. But, on the one hand, the electronic connection with the hospital's information system (computerized medical record) avoids code mistakes and, on the other hand, the daily work of the data managers makes it possible to achieve the completeness of the clinical and epidemiological relevant information. To answer this comment we have expanded the limitations section with a text marked in yellow in the manuscript.

We have also reviewed the reference list and corrected a mistake that was present.

Reviewer 2 Report

Comments and Suggestions for Authors

This is an exciting paper; its topic is essential for the medical profession.

The introduction is well-written. The material and methods section needs improvement in the statistical analysis section.

Is negative binomial regression the suitable model? Does it fit well in terms of the statistical criteria?  You need to explain the role of the permutation test and the sensitivity analysis and tell us how this procedure produces better models.

What is the role of the variable for all cancers in Figure 4? Is this a constant? What is its meaning?

Generally speaking, medical professionals should explain and understand the models.

Do you think cancer specialists can understand your presentations on the statistical models?

Comments on the Quality of English Language

Possible minor corrections 

Author Response

This is an exciting paper; its topic is essential for the medical profession.

The introduction is well-written. The material and methods section needs improvement in the statistical analysis section.

We want to thank the referee for the time devoted to review our manuscript, and for the comments, which have improved the manuscript. In the following paragraphs, we provide a point-by-point response to referee comments.

Is negative binomial regression the suitable model? Does it fit well in terms of the statistical criteria?  You need to explain the role of the permutation test and the sensitivity analysis and tell us how this procedure produces better models.

In this manuscript we analyzed count data. The most common regression models to analyze count data are Poisson and negative binomial regression. Poisson regression assumes no overdispersion. However, there was overdispersion in our data, and in this scenario, negative binomial models fit better. In addition, in the negative binomial models we fit, the assumptions of linearity in model parameters and independence of individual observations were met. To check linearity we plotted the expected counts across the range of years. While to assess independence we computed the Durbin-Watson test. We have added this information in the methods section of the manuscript.

The permutation test is used for testing which of two models with increasing number of joinpoints fits the data better. To test this a ratio is calculated between the sum of squared errors from the null model (no or less joinpoints) and the sum of squared errors from the alternative model (with joinpoints). If the ratio is close to 1, models are similar, if the ratio is large the alternative model is better. The permutation method is used to define how large the ratio has to be for identification as statistically significant. In this method, the residuals from the null model are permuted, and permutation data sets are created. Then, the described ratios are calculated in the permuted datasets. And the following proportion (p-value) is calculated: proportion of permutation datasets in which the ratio values are as extreme as those in the original dataset. If the model with no joinpoints is correct, half of the ratios from the permutation datasets would be larger than the ratio from the original data. On the other hand, if the model with a joinpoint is correct, most of the ratios from the permutation datasets would be lower than the ratio from the original data, and the p-value would be small. The permutation test produces better models because it provides a way of testing if the model with more or less joinpoints is better. We have included a summary of the permutation test explanation in the methods section of the manuscript, and a large description in Supplementary data.

In the sensitivity analyses, we used a second method to estimate joinpoints and we assessed the impact of autocorrelation. We used 2 methods to estimate joinpoints to provide more robust results and not to rely only in 1 method. Both methods provided the same or very close joinpoints. Regarding autocorrelation, we did this sensitivity analysis because the models to identify joinpoints assume uncorrelated errors. To check this assumption we performed additional analysis with correlated errors as defined in the literature. The sensitivity analyses increase the robustness of the results because we can be more confident on the identified joinpoints.    

All the changes are marked in yellow in the manuscript. 

What is the role of the variable for all cancers in Figure 4? Is this a constant? What is its meaning?

In Figure 4, now Figure 3, All cancers refers to the incidence and 1-year mortality of all cancer events independently of cancer localization.

Generally speaking, medical professionals should explain and understand the models.

We agree with the referee.

Do you think cancer specialists can understand your presentations on the statistical models?

This manuscript has been done by medical professionals, including cancer specialists, epidemiologists, and professors of a university medical school. These professionals have made the manuscript as understandable as possible for the medical community.

We agree that the statistical analysis may be complex, but it was the analysis needed to fulfill the manuscript objectives. On the one hand, we provided a simple explanation of the methods section so it would be easier to understand. On the other hand, we included key terms that medical professionals should know, to understand the methods, such as regression models, coefficients, confidence intervals, and sensitivity analyses. Probably, the most complex part in the methods is the one about joinpoint regression, this is why we provided more detail about how the analysis was done. In addition, we provided the functions and programs used in case the medical professionals are familiar with statistical analysis. Although we have done our best to make the manuscript understandable, we realize that it can be improved. Thus, in this revised version of the manuscript, we have included more information in the methods section on the manuscript as well as in the Supplementary data. 

Regarding the results, we think that medical professionals can understand them as these professionals are used to incidence and 1-year mortality figures. To make results easier to interpret we presented cancer trends from regression models in a forest plot, which is a graph used in meta-analyses, and thus, common in articles for the medical community. However, as medical professionals are more familiar with incidence and mortality figures than with outputs of regression models, we have provided a larger description of incidence and mortality standardized rates, in the revised version of the manuscript. In addition, as there was much variation in incidence and mortality data, which could make it difficult to understand the results, we have modified the graphs and included new ones.        

We have also reviewed the reference list and corrected a mistake that was present.

All the changes are marked in yellow in the manuscript.

Reviewer 3 Report

Comments and Suggestions for Authors

Authors have nicely discussed the possible reasons of decline of some types of cancer during the COVID-19 pandemic. However, they use a misleading subtitle as “Effect of the COVID-19 pandemic on cancer incidence and mortality”. This could be better expressed as Incidence of cancer incidence and mortality during the covid-19 pandemic”. Because as authors discuss themselves, “during March-June 2020 there were huge reductions in cancer referrals, gold standard tests, chemotherapy attendance, and patients receiving surgery. Also, in addition, cancer screening was lower after the lockdown compared to before the pandemic period. The reduction in diagnostic tests suggests that a lower number of individuals received a cancer diagnosis during 2020. This reduction in cancer diagnosis together with the decrease of cancer treatments could increase cancer mortality from 2020 on”. Therefore, this observation is not a direct “causal effect” of COVID-19 pandemic on cancer incidence and mortality; it was only an indirect and “non-causal observation”, due mostly to the changes in medical diagnosis/treatment and referrals, etc.  

Author Response

Authors have nicely discussed the possible reasons of decline of some types of cancer during the COVID-19 pandemic. However, they use a misleading subtitle as “Effect of the COVID-19 pandemic on cancer incidence and mortality”. This could be better expressed as “Incidence of cancer incidence and mortality during the Covid-19 pandemic”. Because as authors discuss themselves, “during March-June 2020 there were huge reductions in cancer referrals, gold standard tests, chemotherapy attendance, and patients receiving surgery. Also, in addition, cancer screening was lower after the lockdown compared to before the pandemic period. The reduction in diagnostic tests suggests that a lower number of individuals received a cancer diagnosis during 2020. This reduction in cancer diagnosis together with the decrease of cancer treatments could increase cancer mortality from 2020 on”. Therefore, this observation is not a direct “causal effect” of COVID-19 pandemic on cancer incidence and mortality; it was only an indirect and “non-causal observation”, due mostly to the changes in medical diagnosis/treatment and referrals, etc. 

We totally agree with the referee comment. At no point do the authors attempt to establish a direct causal relationship between the 2020 lockdown due to the Covid-19 pandemic and its impact on cancer incidence or mortality in 2020 and beyond. We have modified the name of the sections 3.5 and 4.3 in the sense the reviewer suggests.

We have also reviewed the reference list and corrected a mistake that was present.

Round 2

Reviewer 1 Report

Comments and Suggestions for Authors

ok